# RE-YOLOv5: Enhancing Occluded Road Object Detection via Visual Receptive Field Improvements

**DOI:** 10.3390/s25082518

**Published:** 2025-04-17

**Authors:** Tianyu Li, Xuanrui Xiong, Yuan Zhang, Xiaolin Fan, Yushu Zhang, Haihong Huang, Dan Hu, Mengting He, Zhanjun Liu

**Affiliations:** 1School of Communications and Information Engineering, Chongqing University of Posts and Telecommunications, Chongqing 400065, China; s220101076@stu.cqupt.edu.cn (T.L.); xiongxr@cqupt.edu.cn (X.X.); s220101027@stu.cqupt.edu.cn (X.F.); s220132224@stu.cqupt.edu.cn (Y.Z.); s220131027@stu.cqupt.edu.cn (H.H.); s220132056@stu.cqupt.edu.cn (D.H.); s220131023@stu.cqupt.edu.cn (M.H.); liuzj@cqupt.edu.cn (Z.L.); 2School of Computer and Internet of Things, Chongqing Institute of Engineering, Chongqing 400056, China

**Keywords:** object detection, occlusion object detection, receptive field, YOLOv5

## Abstract

Road object detection technology is a key technology to achieve intelligent assisted driving. The complexity and variability of real-world road environments make the detection of densely occluded objects more challenging in autonomous driving scenarios. This study proposes an occluded object detection algorithm, RE-YOLOv5, based on receptive field enhancement to assist with the difficult identification of occluded objects in complex road environments. To efficiently extract irregular features, such as object deformation and truncation in occluded scenes, deformable convolution is employed to enhance the feature extraction network. Additionally, a receptive field enhancement module is designed using atrous convolution to capture multi-scale contextual information and better understand the relationship between occluded objects and their surrounding environment. Considering that the ordinary non-maximum suppression method in dense occlusion scenarios will incorrectly suppress the prediction box of the occluded object, EIOU was used to optimize the non-maximum suppression method. Experiments were conducted on two benchmark datasets, KITTI and CityPersons. The proposed method achieves a mean average precision (mAP) of 82.04% on KITTI, representing an improvement of 2.34% over the baseline model. For heavily occluded objects on CityPersons, the Log Average Miss Rate (${\mathrm{MR}}^{-2}$) is reduced to 40.31%, which is a decrease of 9.65% compared to the baseline. These results demonstrate that the proposed method significantly outperforms other comparative algorithms in detecting occluded objects across both datasets.

## 1. Introduction

Currently, Advanced Driver Assistance Systems (ADASs) have been widely applied in automobiles. With their gradually improving application capabilities and operational authority, intelligent assisted driving has become a crucial step toward achieving true autonomous driving [1]. ADAS combines computer vision technology, machine learning technology, and perception technology. It performs real-time monitoring of the vehicle’s surroundings using advanced on-board sensors such as LiDAR and monocular or binocular cameras. These sensors collect environmental information and input it into computing devices. The vehicle’s intelligent algorithms then analyze traffic factors to provide timely feedback through output devices, enhancing safety by identifying hazards and intervening in dangerous situations [2]. As a prerequisite for effective ADAS operation, environmental sensing is fundamental to vehicle decision making and control.

The most common vehicle environment sensors are vision-based. These sensors, which closely mimic human visual perception, can independently perform environmental sensing tasks at low cost. Recent developments in hardware acceleration technology and deep learning-based object detection algorithms have further solidified their technical foundation.

Road object detection, essential for intelligent and autonomous driving, involves identifying vehicles, pedestrians, cyclists, and traffic signs. Accurate detection provides reliable decision-making support for vehicle control systems, ensuring safe driving. This makes road object detection a critical research priority [3]. Recent studies highlight that robust road perception systems not only enhance driving safety but also provide critical data infrastructure for predictive road maintenance analytics. As demonstrated in comprehensive reviews of pavement management systems, machine learning-driven detection frameworks can simultaneously monitor real-time traffic objects and quantify long-term road surface degradation patterns—a dual capability crucial for both vehicle control and infrastructure preservation [4]. However, real-world road scenes are complex and variable. Key challenges include environmental interference from adverse weather or poor lighting as well as widespread occlusion. Occluded objects—whether obscured by backgrounds or other objects—often exhibit deformed or truncated features that general detectors struggle to recognize.

In this paper, we address the challenge of detecting occluded objects in road scenes, which are often characterized by truncation, deformation, and other features that make them difficult for general detectors to identify. Leveraging deep learning-based object detection technology, we investigate road object detection algorithms in intelligent assisted driving scenarios and design an efficient neural network model for the accurate detection of occluded objects in road environments. The main contributions are as follows.

(1)An object detection algorithm RE-YOLOv5 based on receptive field enhancement is proposed, which effectively solves the problem of low accuracy in occluded object detection caused by the difficulty in extracting effective features in complex road scenes.(2)To improve the feature extraction ability of the backbone network for occluded objects, the CSPLayer with Deformable Convnets (CSPD) module is designed by introducing deformable convolution in the CSPLayer. The Receptive Field Atrous Enhancement (RFAE) module is designed using atrous convolution to obtain richer contextual information.(3)In the post-processing stage, the network is optimized using an EIOU-NMS algorithm, which is used to improve the accuracy of occluded object detection. Various experiments validate the superiority of this method.

The rest of the paper is organized as follows. Section 2 reviews the methods for occlusion object detection. Section 3 proposes the network structure and optimizes the non-maximal value suppression algorithm. Experimental results and analyses are given in Section 4. Finally, the paper is summarized in Section 5.

## 2. Related Work

### 2.1. Occlusion Object Detection

In traffic road scenes, the phenomenon of objects being occluded is widespread. For example, dense pedestrians and dense traffic are typical scenarios where occlusion occurs. The types of occlusion can be generally divided into two categories: inter-class occlusion and intra-class occlusion [5]. Inter-class occlusion occurs between different object categories, such as when background elements obscure objects of interest or when distinct object types overlap. Intra-class occlusion refers to mutual obstruction among objects of the same category, such as multiple pedestrians in a crowd.

Inter-class occlusion can cause changes in object shape and obscure critical features, leading to incomplete feature extraction and missed detections. Intra-class occlusion causes prediction boxes of adjacent objects to cluster closely, which are easily suppressed by standard non-maximum suppression (NMS) algorithms, thereby increasing missed detection rates. To address these challenges, researchers have developed targeted solutions.

Kim et al. [6] proposed BBCNet (Bounding-Box Critic Network), which is a plug-in network that uses bounding-box estimators to identify occlusion regions. Through adversarial learning, this approach forces detectors to learn occlusion features. When integrated into Faster R-CNN and SSD frameworks during training and inference, BBCNet significantly improves the detection of occluded vehicles. Mao et al. [7] enhanced YOLOv3 for overlapping object detection in road scenes by redesigning its post-processing stage. Their flexible NMS algorithm, validated on autonomous driving datasets, demonstrates improved performance. Tian et al. [8] developed SA-YOLOv3, which optimizes regression loss functions to enhance the localization of occluded objects. For pedestrian detection under occlusion, Tan et al. [9] designed the Prior-based Receptive Field Block (PRFB). This module aligns receptive fields with pedestrian aspect ratios, effectively reducing background interference during multi-scale feature extraction. Xie et al. [10] introduced a deformable attention-guided network for pedestrian detection. By combining deformable convolutional structures with attention mechanisms, their method samples features from non-rigid locations and aggregates global context through attention maps. Addressing feature degradation in small occluded objects, Liu et al. [11] proposed SE-YOLOv4. This algorithm replaces linear interpolation upsampling with PixelShuffle-based path aggregation and pyramidal convolutional attention, preserving feature integrity through channel-wise periodic shuffling.

### 2.2. Receptive Field Enhancement

The receptive field defines the spatial region of an input image that influences a neuron’s activation. A larger receptive field enables models to capture global contextual patterns, while a smaller one preserves fine-grained local details. In computer vision, multi-scale receptive field design allows models to jointly model object–environment relationships, thereby improving detection robustness. For occluded object detection, where partial object features are missing, leveraging contextual information through receptive field adaptation becomes critical. This compensates for incomplete features by inferring relationships between visible object parts and their surroundings.

Recent studies have explored various receptive field enhancement strategies for occlusion handling: Chi et al. [12] developed a selective refinement network with multi-branch modules to adaptively fuse features from different receptive fields. Najibi et al. [13] proposed context integration through enlarged convolutional windows around candidate regions, demonstrating that simple architectural changes can improve occlusion reasoning. Wang et al. [14] introduced large-kernel convolutions combined with channel-wise attention mechanisms, simultaneously expanding receptive fields and enhancing directional feature sensitivity. Deng et al. [15] implemented context modeling by deploying independent modules across five feature pyramid levels, enabling multi-scale rigid context fusion. Yu et al. [16] specifically addressed face occlusion through a receptive field enhancement module and designed normalized Wasserstein distance (NWD) loss to mitigate the localization bias for small occluded objects. These works collectively validate that context-aware receptive field design effectively mitigates occlusion challenges, establishing it as a promising research direction.

### 2.3. Non-Maximum Suppression

In object detection, the non-maximum suppression (NMS) algorithm is typically applied before the network’s final output to select optimal prediction boxes from overlapping candidates. The core principle of NMS involves first ranking all prediction boxes by their confidence scores. The box with the highest confidence is selected as the base, and remaining boxes are iteratively compared against it. If a box’s Intersection-Over-Union (IOU) with the base exceeds a predefined threshold, its confidence is set to zero (suppressed); otherwise, it is retained. This process repeats until all boxes are evaluated.

Most current detection algorithms rely on Greedy-NMS [17], which determines suppression solely through IOU calculations. However, Greedy-NMS overlooks the spatial relationships between boxes. When neighboring boxes surpass the IOU threshold, the algorithm abruptly discards them by zeroing their scores. This aggressive suppression increases missed detections, particularly in occlusion-prone scenarios.

To mitigate this issue, Soft-NMS [18] introduces a graduated penalty mechanism. Instead of outright suppression, it reduces the confidence scores of high-overlap boxes proportionally to their IOU values. Boxes with substantial overlap receive stronger penalties, while distant boxes remain unaffected. Despite this improvement, Soft-NMS may still penalize well-localized boxes with low confidence more harshly than poorly-localized boxes with high confidence—a problematic scenario when localization and classification confidence are mismatched.

Further advancing this field, DIOU-NMS [19] incorporates centroid distance metrics into the suppression criteria via Distance-IOU (DIOU). By jointly considering the overlap area and centroid proximity, the method more effectively resolves conflicts between overlapping predictions. Nevertheless, DIOU-NMS does not account for differences in box aspect ratios, leaving room for improvement in handling objects with significant shape variations.

### 2.4. Emerging Research Directions

In recent years, multimodal fusion and self-supervised pre-training have emerged as pivotal research directions in intelligent transportation detection. Cross-modal collaborative detection methods have demonstrated significant advantages. Lu et al. [20] proposed a Shared-Specific Feature Transfer Framework, which decouples modality-shared features from modality-specific ones, achieving a 22.5% mAP improvement on the SYSU-MM01 dataset. Similarly, Labbaki et al. [21] innovatively designed an Orthogonal Sequential Fusion Mechanism that mitigates feature redundancy through phased fusion and orthogonal constraints, outperforming traditional parallel fusion approaches. These studies underscore the critical role of inter-modal complementarity mining and structured fusion strategies in enhancing detection robustness.

Self-supervised learning has also achieved breakthroughs in unlabeled data utilization and domain adaptation. Zhang et al. [22] introduced a Pseudo-Label Cross-Teaching Framework for scene adaptation, leveraging background invariance enhancement and hybrid data augmentation to significantly improve detection accuracy in fixed-view scenarios. In monocular depth estimation, Saunders et al. [23] developed a Direction-Aware Convolutional Network that addresses orientation sensitivity through environmental context accumulation, achieving state-of-the-art performance on benchmarks like KITTI. Concurrently, Han et al. [24] constructed a Weather-Augmented Robustness Framework using pseudo-supervised losses and weather-aware augmentation, surpassing existing methods in foggy/nighttime scenarios. These advancements highlight the potential of self-supervised learning for generalization in complex transportation environments.

To enable a systematic analysis of advancements and limitations in occlusion handling for object detection, we provide a structured summary of existing methodologies in Table 1. As illustrated, each approach employs distinct technical strategies to address specific challenges in this domain. Although the above method improves the detection accuracy of occluded objects to a certain extent, there are still many problems, such as the more serious situations of missed detection and misdetection. To address these issues, this paper designs a method of receptive field enhancement and uses this method to experimentally validate and analyze the method with YOLOv5 as the baseline model.

## 3. System Model

### 3.1. RE-YOLOv5 Overall Structure

The overall network framework of the detection model RE-YOLOv5 in this paper is based on YOLOv5, as shown in Figure 1. In the backbone network, we replace a portion of the standard convolutions in YOLOv5’s CSPLayer with deformable convolutions better suited for occluded object feature extraction. Each modified CSPLayer is renamed as CSPD (CSPLayer with Deformable Convolutions, CSPD). To enhance the network’s contextual information capture capability, Receptive Field Augmentation Modules (RFAMs) are introduced to the last three feature layers of the feature extraction network for multi-scale context modeling. Finally, prior to detection output, an EIOU-based non-maximum suppression (NMS) algorithm is employed to suppress redundant prediction boxes. The following subsections will elaborate on the aforementioned methodology, with Table 2 providing the notation definitions used in subsequent subsections.

### 3.2. CSPLayer with Deformable Convnets

In order to enhance the feature extraction ability of the model for occluded objects, this study uses a 3 × 3 sized DCNv2 to replace some of the convolutional layers. Deformable Convnets v2 (DCNv2) [25] is a new type of convolution operation, which aims to improve the CNN’s ability to adapt to changes in object shape. The shape of the convolution kernel is fixed in ordinary convolution operations, while the shape and position of the DCNv2 convolution kernel can be automatically adjusted according to the differences in its object shape. In road occlusion environments, there is extensive occlusion between objects, and thus their effective features are characterized by multiple states. In addition, the pose of the pedestrian object in the road changes at any time, and it is difficult to extract features effectively by ordinary convolution. Therefore, in road occlusion environments, DCNv2 has a better feature extraction capability than ordinary convolution.

As illustrated in Figure 2, the CSPDLayer of RE-YOLOv5 comprises a combination of a CBS component and a DCNv2-Residual Unit. The CBS’s convolutional layers, all featuring 1 × 1 convolutional kernels, primarily control the channel relationships between layers. Meanwhile, the DCNv2-Res Unit, serving as the smallest residual unit within the CSPDLayer, is obtained by cascading a CBS with a Direction-wise Batch Splitting (DBS) and then connecting the residuals. In this structure, the DCNv2 within the DBS is of size 3 × 3. This size is chosen because it provides a sufficient receptive field, making it well suited for feature extraction in the backbone network.

### 3.3. Receptive Field Atrous Enhancement Module

In occluded object detection, the presence of occlusion prevents the effective extraction of some object information, creating challenges for detection. Utilizing contextual information effectively helps the model learn the relationship between the visible parts of the object and the surrounding environment, thereby improving the detection performance for occluded objects. In this paper, a Receptive Field Atrous Enhancement (RFAE) module based on atrous convolution (AC) is designed. The structure of RFAE is illustrated in Figure 3.

Atrous convolution (AC) [26] alters the size of the convolution kernel by introducing voids in the middle of the kernel. This modification allows for the acquisition of different receptive fields and corresponding contextual information. In contrast to regular convolution, AC’s feature extraction process no longer involves convolving the convolution kernel with neighboring pixels for every element. Instead, it skips certain pixels based on a specified dilation ratio before performing convolution with the input pixels. Although the size of the receptive field can be controlled by adjusting the kernel size and stride, increasing the kernel size introduces additional computational overhead, while increasing the stride results in sparser feature extraction. In contrast, AC allows for the extraction of rich contextual information using the same size of the convolutional kernel without causing excessive sparsity in the extracted features.

The RFAE consists of three parts. The first part consists of four groups of AC; here, the RFAE sets four void rates of AC that are used to provide multi-scale receptive field information for the feature map, while the number of each group of AC is set to 1/4 of the input feature map, and the size of the convolution kernel *k* is 3. In the RFAE, the dilated ratios are set to 1, 2, 3, and 4, at which time the equivalent receptive field size can be obtained according to Equation (1).(1)RF=d×(k−1)+1
in the above equation, RF is the receptive field size, *d* is the dilation ratio, and *k* denotes the size of the convolution kernel.

Since the size of the convolution kernel used is 3 × 3, the receptive field sizes are 3, 5, 7 and 9 according to Equation (1). In addition, in order to reduce the information loss caused by the voids of the atrous convolution, RFAE contacts the input feature maps with the feature maps of the AC outputs in the channel direction, which retains the original information and at the same time introduces the information of the receptive fields of different sizes. Assuming that the input feature map is denoted by *F*, the output feature map F1 after the above process can be described as shown in Equation (2).(2)$$F1=\mathrm{Concat}(\left[F,{\mathrm{AC}}_{d=1}(F),{\mathrm{AC}}_{d=2}(F),{\mathrm{AC}}_{d=3}(F),{\mathrm{AC}}_{d=4}\left(F\right)\right])|$$
the second part uses the Efficient Channel Attention (ECA) [27] mechanism to enhance the feature map channel correlations in the first part and suppress useless channel information to some extent. ECA first compresses the feature map in the spatial dimension using global average pooling (GAP), obtaining a set of feature information related to the channel dimension. Then, adaptive one-dimensional convolution with convolution kernel size *k* is used to generate the channel weights, which are channel-weighted to the input feature map after Sigmoid normalization. Assuming that the output feature map of the above process is F2, it can be described as shown in Equation (3).(3)$$F2=\mathrm{Sigmoid}({\mathrm{AConv}}^{k}\left(\mathrm{GAP}\left(FI\right)\right))⊗FI$$
where ${AConv}^{k}$ denotes adaptive one-dimensional convolution with convolution kernel size *k*. GAP denotes the global average pooling of the feature map, and since the number of feature map channels in the network is an integer power of 2, the value of *k* can be determined by Equation (4).(4)$$k=\psi \left(C\right)={\left|\frac{\log_{2}C+b}{\gamma }\right|}_{odd}$$
In the above equation, *C* denotes the number of channels in the input feature map. ${∥}_{odd}$ denotes the nearest odd number taken when it is not possible to divide γ, and the values of γ and *b* are taken by default to be 2 and 1. Longer interactions are guaranteed for high-dimensional channels, and shorter interactions are guaranteed for low-dimensional channels by the mapping ψ(·). The mapping is based on the fact that the channels have longer interactions, and the lower-dimensional channels are able to perform shorter interactions.

The last part of the RFAE is to update the number of channels of the input feature map using a convolutional layer of 1 × 1 size, which keeps the number of channels of the input and output feature maps of the RFAE unchanged so that the number of channels of the feature layer will not be changed when the RFAE is integrated into the original model, and the final feature map $F^{\prime }$ output from the RFAE is denoted by Equation (5).(5)$$F^{\prime }={\mathrm{Conv}}^{l\times l}\left(F2\right)$$

### 3.4. EIOU-NMS: Enhanced Post-Processing

In this paper, EIOU (Efficient IOU) [28] is used as a method for updating the confidence level of NMS, which is called EIOU-NMS. EIOU-NMS takes into account the overlapping area of the prediction boxes, the center distance of the prediction boxes, and the difference between the width and height of the prediction boxes. The EIOU-NMS algorithm is shown in Algorithm 1.
**Algorithm 1** EIOU-NMS.**Input:** $B=\left\{b_{1},b_{2},\ldots ,b_{n}\right\},S=\left\{S_{1},S_{2},\ldots ,S_{n}\right\},N_{t}$B: List with all prediction boxesS: Confidence values corresponding to all prediction boxes$N_{t}$: Confidence threshold**Output:**   The final list of prediction boxes D    **Implementation process:****Step1:** Initialize the prediction box output list D=Φ**Step2:** Sort *S* and set the prediction box with the highest confidence as the base box M**Step3:** Take M from B and put it into D**Step4:** Iterate through list B and remove the extra prediction boxes using the following$\mathrm{IOU}\left(M,b_{i}\right)-R_{EIOU}\left(M,b_{i}\right)\geq N_{t}$, then$B=B-b_{i},S=S-S_{i}$ end**Step5:** Repeat steps 1 through 4 to output the list of prediction boxes D when B is empty

The confidence updating formula optimized by EIOU is denoted as Equation (Equation 6).(6)$$S_{i}=\left\{\begin{array}{cc} S_{i}, & \mathrm{IOU}\left(M,b_{i}\right)-R_{EIOU}\left(M,b_{i}\right)<N_{t}\\ 0, & \mathrm{IOU}\left(M,b_{i}\right)-R_{EIOU}\left(M,b_{i}\right)\geq N_{t} \end{array}\right.$$
In the above equation, $S_{i}$ is the updated confidence value, $R_{EIOU}$ is used to measure the spatial positional relationship between the two boxes, and its expression is shown in Equation (Equation 7).(7)$$R_{EIOU}\left(M,b_{i}\right)=\frac{\rho^{2}\left(M,b_{i}\right)}{c^{2}}+\frac{\rho^{2}\left(w,w_{i}\right)}{c_{w}^{2}}+\frac{\rho^{2}\left(h,h_{i}\right)}{c_{h}^{2}}$$
where *w* is the width of the baseline prediction box, *h* is the height of the baseline prediction box, $w_{i}$ is the width of the remaining prediction box, $h_{i}$ is the height of the remaining prediction box, *c* denotes the diagonal length of the smallest outer bounding rectangle of the two prediction boxes, and $c_{w}$ and $c_{h}$ denote the width and height of the smallest outer bounding rectangle, respectively. Figure 4 shows a schematic illustration of EIOU.

Compared with the traditional NMS, using EIOU to update the confidence level allows NMS to measure the positional relationship between two prediction boxes more comprehensively in the process of suppressing redundant prediction boxes, and thus it is more suitable for object detection in dense occlusion scenarios. Algorithm 1 shows the steps of the EIOU-NMS algorithm execution.

## 4. Experiments and Discussion

### 4.1. Experimental Setup

#### 4.1.1. Datasets and Data Preprocessing

In this paper, we use KITTI [29] and CityPersons [30] datasets for occlusion object detection experiments. The KITTI dataset contains nine categories of labeled information: Car, Van, Truck, Pedestrian, Person-sitting, Cyclist, Tram, Misc, and DontCare, with DontCare denoting unlabeled regions and Misc as irrelevant miscellaneous items. Its training set contains 7481 images, the test set includes 7518 images, and the test set provides no labeling information. In this paper, the experimental setup removes the unlabeled test set, mixes the training set with the validation set, and then reclassifies the training set, validation set, and test set as 8:1:1. Meanwhile, the categories of the KITTI dataset are reclassified: the four categories of Car, Van, Truck, and Tram are merged into the category of Car; Pedestrain and Person-sitting are merged into Pedestrian; and Pedestrain and Person-sitting are merged into Pedestrian. The Cyclist category remained unchanged, and the other irrelevant categories were removed. The percentage of cars, pedestrians, and cyclists that are occluded in the KITTI dataset is 53.4%,40.6%, and 44.5%, respectively. The number of instances in the KITTI dataset as well as the number of instances that are occluded are given in Table 3.

Moreover, the KITTI dataset also grades the detection difficulty of the dataset according to the occlusion rate of the object and the size of the object, and its detection difficulty can be categorized into easy, moderate, and hard. Table 4 shows the basis of the KITTI dataset’s judgment for detection difficulty.

The CityPersons dataset is used for the study of pedestrian detection in autonomous driving scenarios with annotations provided exclusively for the pedestrian class. The dataset consists of 2975 images for training, 500 for model validation, and 1575 for testing. Notably, the test set is unlabeled. In the CityPersons dataset, pedestrians with over 65% visibility are labeled as “Reasonable” (i.e., largely visible), those with 65–90% visibility are labeled as “Partial occlusion” (partially obscured), and those with 20–65% visibility are labeled as “Heavy occlusion” (severely obscured). Pedestrians with no occlusion are labeled as “Bare”. In this paper, 2975 images are used for training, quantitative analysis is conducted on a validation set of 500 images, and visualization is performed on the test set.

In the perspective of data preprocessing, considering the specificity of the occluded scene, this paper uses the Mixup [31] data enhancement method. The idea of the implementation of the Mixup enhancement method is to fuse two scenes into one scene. Before the network training, two samples will be randomly selected from each batch of training samples, and then the information in the images will be blended in a certain ratio to generate a new image. The Mixup method simulates the complex background of the image and also simulates the occluded state to some extent for the objects to be detected, thus indirectly providing the detector with more learning samples of the occluded objects.

#### 4.1.2. Experimental Parameter Settings

The experimental configuration was set as follows: all models were trained for 300 epochs using the Stochastic Gradient Descent (SGD) optimizer with an initial learning rate of 0.01. The implementation leveraged an NVIDIA GeForce RTX 3090 GPU (24 GB VRAM) with CUDA 11.3 acceleration, which was accompanied by an Intel^®^ Core™ i5-12500 CPU. The software stack included PyTorch 2.0.1 and Python 3.9. The training method is freeze training, the pre-training weight file is loaded first for freeze training, the parameters of the backbone network are unchanged in the freezing phase, and all the parameters will be involved in the training in the unfreezing phase.

#### 4.1.3. Evaluation Indicators

The selection of performance metrics is rigorously justified based on dataset characteristics and established practices within the research community. For the KITTI dataset, we adopt mean Average Precision (mAP) as it holistically evaluates multi-class detection performance (cars, pedestrians, cyclists) across diverse occlusion levels and object scales. As the de facto standard for autonomous driving benchmarks, mAP effectively balances localization accuracy, via IOU threshold and occlusion tolerance, aligning with the practical requirements of traffic scenario analysis. For the CityPersons dataset, we employ the Log Average Miss Rate (${\mathrm{MR}}^{-2}$) to specifically address pedestrian detection challenges under severe occlusion. This metric, formally endorsed by the benchmark protocol, quantifies algorithmic robustness to partial visibility by emphasizing missed detection rates in densely occluded scenarios. The dual-metric strategy provides complementary evaluations, mAP reflects comprehensive performance improvements, while ${\mathrm{MR}}^{-2}$ validates targeted optimizations for occlusion handling. Together, these metrics demonstrate both generalized accuracy enhancements and domain-specific advancements in safety-critical applications.

### 4.2. Comparative Experimental Analysis

This study first conducts comparative experiments on the KITTI dataset. As shown in Table 5, where “E” denotes Easy, “M” Moderate, and “H” Hard scenarios, the following observations are made. For the Car category, RE-YOLOv5 outperforms the other listed algorithms in general and moderately occluded scenarios, while BBCNet achieves the highest precision of 86.2% under severe occlusion with RE-YOLOv5 ranking second. For Pedestrian detection, RE-YOLOv5 demonstrates superior accuracy with AP values of 86.56%, 75.56%, and 69.01% across the three scenarios, respectively. Similarly, for Cyclist detection, while PSNS-SSD achieves the highest AP of 88.48% in general scenarios with RE-YOLOv5 following closely, RE-YOLOv5 attains higher precision in partially and severely occluded scenarios at 76.44% and 73.13% AP, respectively, surpassing all compared methods. Furthermore, the mean average precision (mAP) of RE-YOLOv5 exceeds that of the classical detection model Faster-RCNN by 19.94%, and it outperforms state-of-the-art road object detection models PSNS-SSD and PointRCNN by 7.63% and 11.93%, respectively. Overall, RE-YOLOv5 demonstrates superior performance in occluded object detection accuracy on the KITTI dataset.

To further validate the detection performance of RE-YOLOv5, experiments were conducted using the CityPersons dataset, and RE-YOLOv5 was also compared with some of the methods in the literature. Table 6 demonstrates the comparison of the results of RE-YOLOv5 with other methods on the CityPersons validation set.

According to Table 6, it can be seen that the detection of RE-YOLOv5 is improved relative to YOLOv5 in all four conditions. In the case of no occlusion (Bare), the MR-2 of YOLOv5 is 7.80%, while that of RE-YOLOv5 is 5.16%, which is a decrease of 2.64%. In the case of a small amount of occlusion (Reasonable), YOLOv5 is 15.24%, while that of RE-YOLOv5 is 10.98%, which is a decrease of 4.26% in the $MR^{-2}$ value. In the case of an occlusion more severe (Partial) scenario, YOLOv5 is 18.19% and RE-YOLOv5 is 13.49%, which is a decrease of 4.7%. For the severe occlusion (Heavy) scenario, YOLOv5 is 49.96% and RE-YOLOv5 is 40.31%, which is the most significant enhancement with a decrease of 9.65%. The synthesis of the above experimental results shows that the RE-YOLOv5 algorithm proposed in this paper is effective.

Additionally, this study compares RE-YOLOv5 with other methods on the CityPersons validation set. For example, MF-CSP utilizes a semantic feature enhancement module to fuse multi-level feature maps for semantic enhancement. MF-CSP demonstrates acceptable detection performance in Bare, Reasonable, and Partial scenarios, but it shows limited effectiveness in Heavy scenarios with an $MR^{-2}$ of 47.9%, while RE-YOLOv5 achieves a 7.59% lower $MR^{-2}$ under the same conditions. As shown in Table 6, MAFA-Net achieves the best performance in Partial scenarios with an $MR^{-2}$ of 10.05%, whereas RE-YOLOv5 exhibits a 3.44% higher $MR^{-2}$. However, under Heavy conditions, MAFA-Net’s $MR^{-2}$ rises to 43.84%, surpassing RE-YOLOv5 by 3.53%. In Bare and Reasonable scenarios, RE-YOLOv5 still maintains lower Miss Rates. Compared to state-of-the-art (SOTA) models in road object detection, FA-Net achieves an $MR^{-2}$ of 11.6% in Reasonable scenarios, which is 6.02% higher than RE-YOLOv5. CCFA-Net attains an $MR^{-2}$ of 5.66% in Bare scenarios yet still underperforms RE-YOLOv5 by 0.5%.

### 4.3. Qualitative Analysis of Occlusion Object Detection

In order to more intuitively compare the detection effect of YOLOv5 as well as in occluded environments, the experiments were carried out by selecting part of the data from the KITTI test set as well as from the CityPersons test set, and in order to better reflect the detection differences between the two algorithms, the regions used for the comparisons have been additionally labeled out in the figure, in which the results of the KITTI dataset on the KITTI dataset are as demonstrated in Figure 5.

Two common occlusion scenarios were selected as examples for the experiment. The first row of Figure 5 shows intra-category occlusion, where cars are closely aligned at both ends of the road, and there is a white car on its left side that is occluded by other vehicles, which is not detected by YOLOv5, but in (b) in the figure, we can see that RE-YOLOv5 can detect it. In the second row of inter-category occlusion, the car in the labeled area in the figure is not only occluded by the car in front of it but also by pedestrians, which is not detected by YOLOv5 in this scenario; however, in figure (b), we can see that RE-YOLOv5 can still detect it. Through the above visualized results, it is easy to find that RE-YOLOv5 has a better detection effect compared to YOLOv5 in the occlusion scenario of the KITTI dataset.

Figure 6 shows a comparison of the actual detection results on the CityPersons test set. In the detection scenario in the first row, there is an obscured pedestrian behind the pram that is not detected, whereas in figure (b), it can be seen that RE-YOLOv5 can be detected; in the second row of the comparison results, there is an obscured pedestrian on the left side that is not detected by YOLOv5, but it can be detected by RE-YOLOv5. Comparing the two results, we can see that RE-YOLOv5 is better at overall detection.

To further validate the real-world occlusion detection performance and hardware compatibility of RE-YOLOv5, we conducted practical testing using dashcam video data. The experiments leveraged the KITTI-trained model deployed on a system with NVIDIA GeForce RTX 2060 (12 GB VRAM) and Intel^®^ Core™ i5-10400F CPU @ 2.90 GHz. The video data were also sourced from publicly available internet repositories with two randomly selected frames undergoing comparative detection analysis. The results are shown in Figure 7, and it can be seen that the FPS of RE-YOLOv5 is also reduced compared to YOLOv5 detection because deformable convolution increases the inference time, but it is better than YOLOv5 for occluded object detection in real driving scenarios.

To visually evaluate the effectiveness of CSPD and RFAE modules, we present feature heatmap visualizations of their outputs on the CityPersons dataset in Figure 8, which includes three columns: (a) input images, (b) CSPD output heatmaps, and (c) RFAE output heatmaps. The visualization reveals strong activations (red regions) in densely occluded object areas, while moderate background responses (yellow regions) indicate a computational trade-off inherent to contextual feature extraction.

### 4.4. Ablation Experiment

In order to verify the effectiveness of the CSPD, RFAE, and EIOU-NMS used in this paper for occluded object detection, ablation experiments were conducted on each module separately. Method 1 is YOLOv5, replacing the CSPLayer in the backbone network. CSPD is recorded as Method 2. After introducing RFAE in YOLOv5, it is recorded as Method 3. Replacing the Greedy-NMS of YOLOv5 with the EIOU-NMS is recorded as Method 4. The use of CSPD and RFAE at the same time is recorded as Method 5. Last, Method 6 is recorded as RE-YOLOv5. The results of its ablation experiments on the CityPersons dataset as well as the KITTI dataset are shown in Table 7 and Table 8.

The ablation results in Table 7 and Table 8 indicate that replacing the CSPLayer with CSPD in YOLOv5’s backbone network yields more significant improvements compared to RFAE and EIOU-NMS, primarily due to the deformable offsets introduced in CSPD’s convolutional process, which enhance feature extraction for occluded objects. However, CSPD increases model parameters by 0.62 MB and reduces FPS by 10. Similarly, RFAE improves detection accuracy but introduces an additional 1.714 MB of parameters and reduces FPS by 6. While EIOU-NMS contributes less to precision than CSPD and RFAE, it adds no extra parameters and minimally impacts FPS. Overall, RE-YOLOv5, despite its higher parameter count, achieves superior accuracy in occlusion scenarios while maintaining real-time detection capability at 43 FPS, which remains sufficient for practical applications.

### 4.5. Comparison of Different Reprocessing Methods

To further validate the post-processing effect of the EIOU-NMS used in this section, experiments were conducted to compare the EIOU-NMS with the Greedy-NMS and the more advanced DIOU-NMS on KITTI as well as CityPersons data, and the results of the experiments are shown in Table 9 and Table 10.

According to Table 9 and Table 10, it can be seen that the overall effect is better when using EIOU-NMS, which verifies that the use of EIOU-NMS can suppress the redundant prediction boxes more effectively and help improve the detection accuracy of the model. The performance improvement is due to the EIOU metric, which makes the suppression criterion not only limited to the overlap region between two prediction boxes and the distance between the centroids but also focuses on the differences in width and height between the boxes. In addition, the EIOU-NMS method can be easily added to different models without additional training.

## 5. Conclusions

In this paper, an occlusion object detection algorithm RE-YOLOv5 based on receptive field enhancement is proposed for the road occlusion object detection problem in intelligent assisted driving scenarios. Firstly, in the perspective of feature extraction, considering that the features of the objects under the occlusion conditions will be truncated and deformed, the use of deformable convolution to replace the ordinary convolution can be more effective in the extraction of the features of the objects in the occlusion environment. In addition, in order to more effectively establish the connection between the occluded object and the surrounding background information, and to enrich the contextual information, this paper designs the receptive field enhancement module through the atrous convolution and channel attention mechanism to obtain the contextual information of multiple scales and improve the detection effect of the occluded object. In the perspective of non-maximum suppression, the problems that Greedy-NMS would have in occluded environment detection are analyzed, and it is proposed to use the more advanced EIOU-NMS for non-maximum suppression of the detection results to prevent overlapping objects from being suppressed incorrectly. Finally, experiments are conducted on the KITTI dataset and the CityPersons dataset, the mAP on KITTI reaches 82.04%, and the Log Average Miss Rate of heavy occluded objects on CityPersons is only 40.31%. The experimental results show that RE-YOLOv5 has better results than the baseline scheme as well as other detection algorithms in occluded object detection.

## Figures and Tables

**Figure 1 sensors-25-02518-f001:**
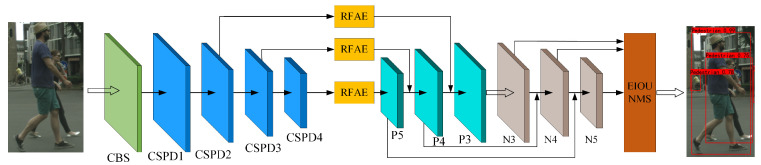
Model overall structure of RE-YOLOv5.

**Figure 2 sensors-25-02518-f002:**
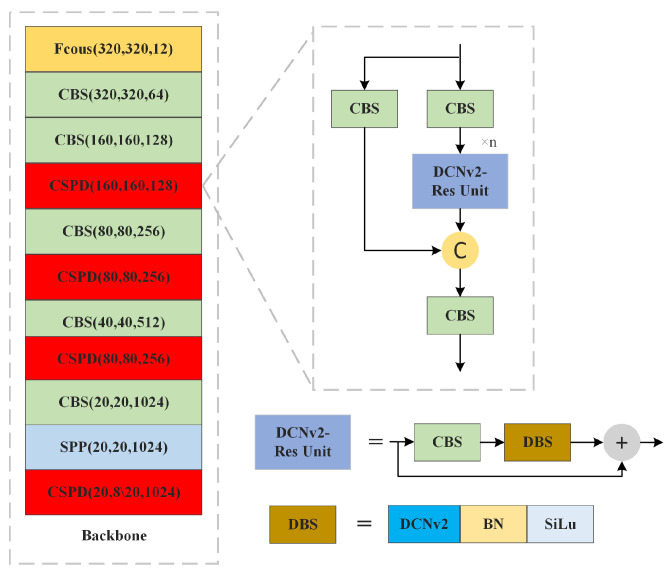
Schematic diagram of backbone network improvement.

**Figure 3 sensors-25-02518-f003:**
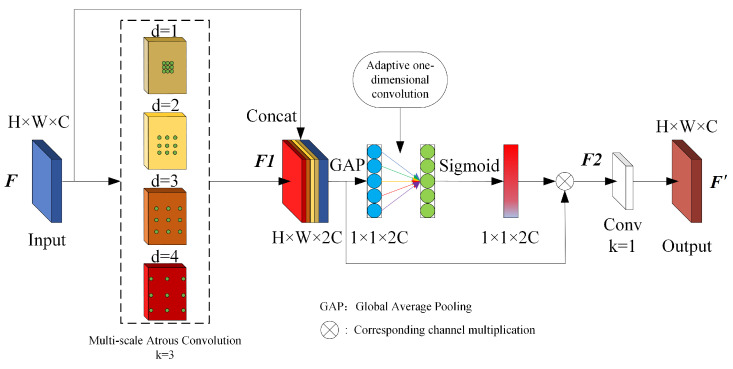
Schematic diagram of RFAE structure.

**Figure 4 sensors-25-02518-f004:**
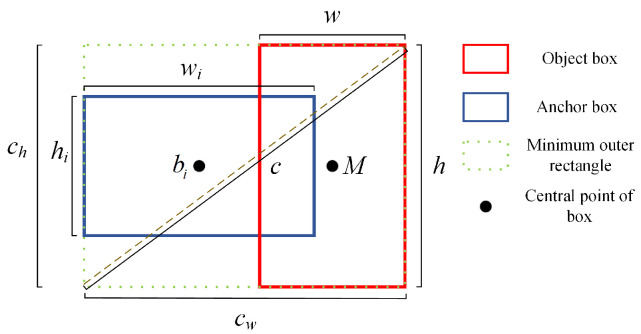
The schematic diagram of EIOU.

**Figure 5 sensors-25-02518-f005:**
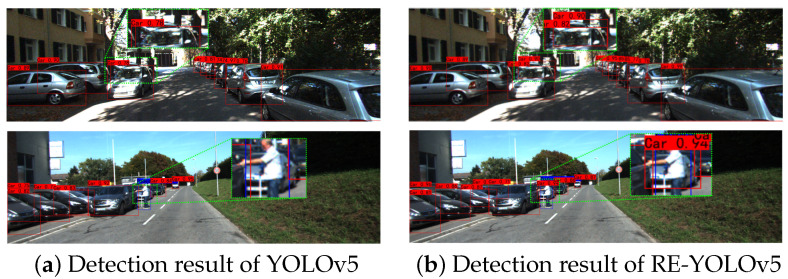
Comparison of KITTI dataset detection results.

**Figure 6 sensors-25-02518-f006:**
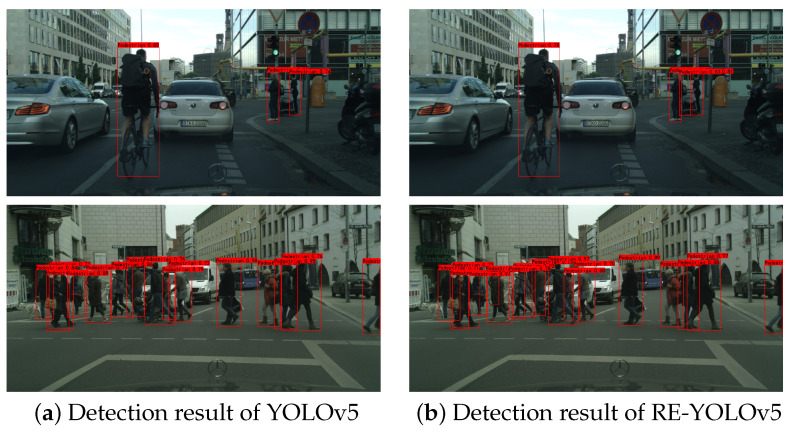
Comparison of CityPersons dataset detection results.

**Figure 7 sensors-25-02518-f007:**
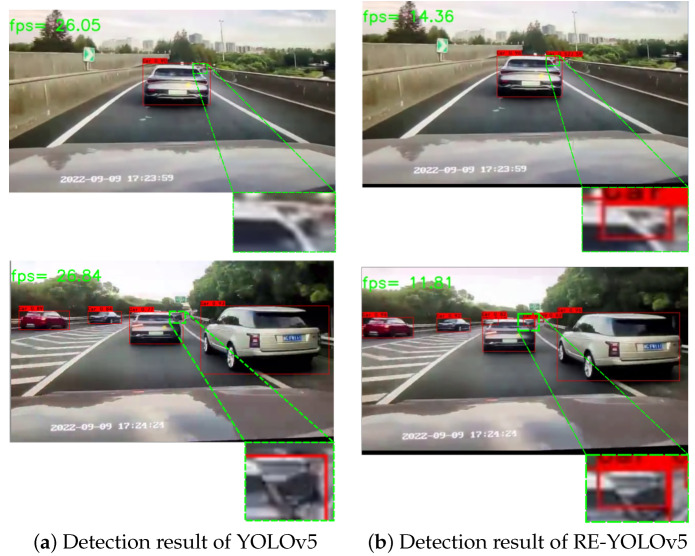
Detection results of occluded objects in video data.

**Figure 8 sensors-25-02518-f008:**
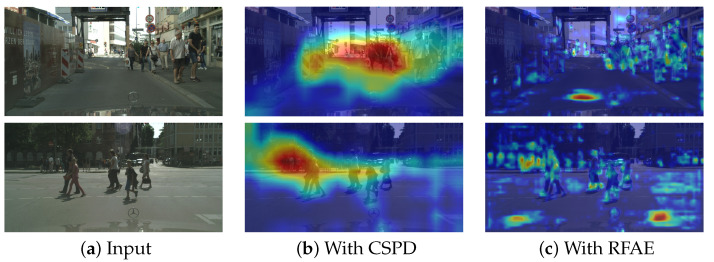
Feature heatmap visualization.

**Table 1 sensors-25-02518-t001:** Comparative analysis of methodologies.

Category	Method	Key Innovation/Advantages	Limitations/Drawbacks
Detection Component-Specific Optimization	BBCNet [6]	Adversarial learning with bounding-box estimators to identify occlusion regions.	Limited to car-to-car occlusion; requires joint training.
	SA-YOLOv3 [8]	Optimized regression loss for occluded object localization.	Focuses on intra-class occlusion only.
Context-Aware Optimization	PRFB [9]	Aligns receptive fields with pedestrian aspect ratios to reduce background interference.	Sensitive to extreme pose variations.
	Deformable Attention-guided Network [10]	Attention-guided deformable convolution for non-rigid feature sampling.	High computational complexity.
	SE-YOLOv4 [11]	PixelShuffle-based upsampling for feature integrity preservation.	Requires careful hyperparameter tuning.
Receptive Field Enhancement	Selective Refinement [12]	Multi-branch adaptive fusion of features from varying receptive fields.	Lacks dynamic adaptation to occlusion patterns.
	Large Kernel Attention [14]	Large receptive field attention for channel-wise features.	May dilute local details in dense scenes.
	NWD Loss [16]	Wasserstein distance for small object localization.	Limited to face detection scenarios.
Post-processing Optimization	Soft-NMS [18]	Gradual confidence reduction based on IOU values.	Fails when location-confidence mismatch exists.
	DIOU-NMS [19]	Incorporates centroid distance into suppression criteria.	Ignores aspect ratio compatibility.
Emerging Research Directions	Shared-Specific Fusion [20]	Decouples modality-shared and specific features (22.5% mAP improvement).	Needs multi-sensor hardware support.
	Orthogonal Fusion [21]	Phased fusion with orthogonal constraints reduces redundancy.	High memory footprint.
	Pseudo-Label Cross-Teaching [22]	Background-invariant adaptation.	Limited to fixed-view scenarios.
	Weather-Augmented Framework [24]	Pseudo-supervised loss + weather-aware augmentation for foggy/nighttime robustness.	Requires weather-specific augmentation.

**Table 2 sensors-25-02518-t002:** List of key notations.

Notation	Description	Notation	Description
AC	Atrous convolution	γ	Adjustment parameter; the default value is set to 2
${AConv}^{k}$	Adaptive one-dimensional convolution	*h*	The height of the baseline prediction box
*B*	List with all prediction boxes	$h_{i}$	The height of the remaining prediction box
*b*	Adjustment parameter, the default value is set to 1	*k*	The size of the convolution kernel
$b_{i}$	Remaining prediction boxes	*M*	Baseline prediction boxes
*C*	The number of channels	$N_{t}$	Confidence threshold
*c*	The diagonal length of the smallest outer bounding rectangle	ψ(·)	Channel mapping
$c_{h}$	The height of the smallest outer bounding rectangle	$R_{EIOU}$	The spatial positional relationship between the two boxes
Concat	Concatenate	RF	Receptive field size
$c_{w}$	The width of the smallest outer bounding rectangle	ρ	Euclidean distance between center points
*D*	The final list of prediction boxes	*S*	Confidence values corresponding to all prediction boxes
*d*	Dilation ratio	$S_{i}$	The updated confidence value
*F*	Feature map	*w*	The width of the baseline prediction box
$F^{\prime }$	Output feature map	$w_{i}$	The width of the remaining prediction box
GAP	Global average pooling	${∥}_{odd}$	Denotes the nearest odd number taken when it is not possible to divide γ

**Table 3 sensors-25-02518-t003:** KITTI dataset occlusion count statistics.

Object Category	Number of Instances	Number of Obscured	Percentage%
Car	28,521	15,231	53.4
Pedestrian	4445	1805	40.6
Cyclist	1612	772	44.5

**Table 4 sensors-25-02518-t004:** Criteria for dividing the detection difficulty of KITTI dataset.

Evaluation Criteria	Easy	Moderate	Hard
Minimum Height of the Bounding Box (Pixel)	40	25	25
Degree of Visibility	Bare	Partial occlusion	Heavy occlusion
Truncation Rate%	15	30	50

**Table 5 sensors-25-02518-t005:** Comparison of different methods on KITTI dataset (%).

Methods	Car			Pedestrian			Cyclist			mAP
	E	M	H	E	M	H	E	M	H	
Faster RCNN	82.9	77.8	66.3	83.3	68.4	62.4	56.4	46.4	42.8	65.1
BBCNet	92.6	91	86.2	86.2	69.5	64.5	75.4	66.7	60.5	76.9
Gaussian-YOLO [32]	89.91	84.36	75.65	58.43	50.95	43.69	45.97	31.37	30.53	56.76
CenterNet	92.3	89.15	82.17	76.53	67.53	59.37	73.48	52.63	50.25	71.49
U-FPNDet [33]	91.3	89.7	81.8	75.7	72.1	68.4	74.5	68.4	62.1	76
CertainNet [34]	93.81	89.36	82.11	76.33	66.13	58.54	78.02	57.49	55.2	72.99
ASSD [35]	89.28	89.95	82.11	69.07	62.49	60.18	75.23	76.16	72.83	75.25
PSNS-SSD [36]	89.13	84.27	78.73	68.69	62.31	57.22	88.48	73.48	68.13	74.41
PointRCNN [37]	93.42	89.11	84.25	55.17	48.10	44.90	81.83	70.94	63.25	70.11
RE-YOLOv5	94.21	92.76	82.65	86.56	75.56	69.01	88.1	76.44	73.13	82.04

**Table 6 sensors-25-02518-t006:** Comparison of different methods on CityPersons verification set ($MR^{-2}$).

Methods	Reasonable (%)	Bare (%)	Partial (%)	Heavy (%)
MF-CSP [38]	11.2	7.2	10.4	47.9
OR-CNN [39]	12.8	6.7	15.3	55.7
MSCM-ANet [40]	11.95	7.9	11.1	50.1
DDFE [41]	12.9	8.2	12.1	50.5
MAFA-Net [42]	11.47	6.29	10.05	43.84
FA-Net [43]	11.6	–	11.9	42.8
FCOS w [44]	13.9	8.18	14.23	47.7
CCFA-Net [45]	–	5.66	13.35	49.99
YOLOv5	15.24	7.8	18.19	49.96
RE-YOLOv5	10.98	5.16	13.49	40.31

**Table 7 sensors-25-02518-t007:** Results of ablation experiments on the CityPersons dataset ($MR^{-2}$).

Methods	CSPD	RFAE	EIOU-NMS	Reasonable (%)	Bare (%)	Partial (%)	Heavy (%)	Parameter (MB)	FPS
1				15.24	7.8	18.19	49.96	47	61
2	✔			13.21	6.26	14.32	45.3	47.62	51
3		✔		14.65	6.88	15.34	46.17	48.714	55
4			✔	14.35	7.03	16.3	47.03	47	58
5	✔	✔		12.88	6.11	14.01	42.61	49.334	49
6	✔	✔	✔	10.98	5.16	13.49	40.31	49.334	43

**Table 8 sensors-25-02518-t008:** Results of ablation experiments on the KITTI dataset (%).

Methods	Car			Pedestrian			Cyclist			mAP
	E	M	H	E	M	H	E	M	H	
1	92.07	89.91	79.31	85.23	74.1	66.71	86.7	73.31	70.01	79.7
2	93.51	91.23	81.37	85.66	75.26	68.35	87.81	75.22	71.98	81.15
3	93.11	91.04	81.22	85.41	74.6	67.66	87.22	74.68	71.46	80.71
4	92.8	90.8	80.11	85.34	74.27	67.21	87.11	74.15	71.25	80.33
5	93.74	92.13	82.22	86.07	75.49	68.56	88.03	75.63	72.58	81.6
6	94.21	92.76	82.65	86.56	75.56	69.01	88.1	76.44	73.13	82.04

**Table 9 sensors-25-02518-t009:** Results of ablation experiments on the KITTI dataset (%).

Methods	Car			Pedestrian			Cyclist			mAP
	E	M	H	E	M	H	E	M	H	
Greedy-NMS	93.74	92.13	82.22	86.07	75.49	68.56	88.03	75.63	72.58	81.6
DIOU-NMS	94.1	92.33	82.46	86.27	75.5	68.66	88.08	76.01	72.88	81.81
EIOU-NMS	94.21	92.76	82.65	86.56	75.56	69.01	88.1	76.44	73.13	82.04

**Table 10 sensors-25-02518-t010:** A comparison of different post-processing methods on CityPersons ($MR^{-2}$).

Methods	Reasonable (%)	Bare (%)	Partial (%)	Heavy (%)
Greedy-NMS	12.88	6.11	14.01	42.61
DIOU-NMS	11.21	5.48	13.64	41.78
EIOU-NMS	10.98	5.16	13.49	40.31

## Data Availability

The data presented in this study are available upon request from the corresponding author.

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
