# Peer review of "RE-YOLOv5: Enhancing Occluded Road Object Detection via Visual Receptive Field Improvements"

_sensors, 2025, doi:10.3390/s25082518_

Round 1
Reviewer 1 Report
Comments and Suggestions for Authors
This paper proposes an occluded object detection algorithm RE-YOLOv5 based on Receptive Field enhancement to solve the problem of difficult identification of occluded objects in complex road environments. There are some comments.
- The object detection method proposed in the paper is based on YOLO. Therefore, its innovation needs further discussion.
- It is recommended that the author search for papers from the top conferences in the field of traffic detection in the past three years and further discuss emerging directions such as multimodal fusion and self-supervised pre-training.
- In the experimental section, the author only compared it with old versions, such as YOLOv5/V7, and did not compare it with specialized models in the transportation field.
- In the experimental section, a single publicly available dataset was used, which did not include complex urban traffic scenarios.
- The introduction of the CBAM module only compares mAP through ablation experiments without analyzing its differential effects on different categories of targets (such as traffic lights and pedestrians).
- It is suggested that the author conduct some necessary discussions on hardware deployment costs and model lightweight (such as pruning), which can guide engineering practice.
Author Response
This paper proposes an occluded object detection algorithm RE-YOLOv5 based on Receptive Field enhancement to solve the problem of difficult identification of occluded objects in complex road environments. There are some comments.
In the PDF file 'Highlighted Manuscript.pdf' within the compressed archive, the modified sections and critical content have been highlighted using color annotations to facilitate review.
Comment 1: The object detection method proposed in the paper is based on YOLO. Therefore, its innovation needs further discussion.
Reply: Thank you for raising this critical question. We have strengthened the discussion of our method's innovation through the following revisions:
In Section 3.1, we explicitly differentiate RE-YOLOv5 from the baseline YOLOv5. The key modifications include: CSPD Layer: Replacing standard convolutions in YOLOv5’s CSPLayer with deformable convolutions (CSPD) to better adapt to occluded object feature extraction; Receptive Field Atrous Enhancement Module: Introducing multi-scale context extraction modules on the last three feature layers of the backbone to enhance contextual awareness; EIOU-NMS: Employing an Enhanced Intersection-over-Union-based non-maximum suppression algorithm before the final output to mitigate erroneous suppression of overlapping prediction boxes. Ablation studies (Tables 6 and 7) quantitatively demonstrate RE-YOLOv5’s improvements over YOLOv5 in both detection accuracy and missed detection rates.
Specifically, our method achieves a 2.34% increase in mAP on the KITTI dataset and a 9.65% reduction in MR-2 on the CityPersons dataset compared to the baseline model. These improvements can be observed in Tables 6 and 7. We appreciate your valuable feedback and welcome further suggestions.
Comment 2: It is recommended that the author search for papers from the top conferences in the field of traffic detection in the past three years and further discuss emerging directions such as multimodal fusion and self-supervised pre-training.
Reply: Thank you for your insightful suggestion regarding emerging directions like multimodal fusion and self-supervised learning. We have systematically reviewed recent literature (CVPR, ICCV, ICLR) from the past three years and added Section 2.4 Emerging Research Directions in the revised manuscript to address these topics:
In recent years, multimodal fusion and self-supervised pre-training have emerged as pivotal research directions in intelligent transportation detection. Cross-modal collaborative detection methods have demonstrated significant advantages. Lu et al. proposed a Shared-Specific Feature Transfer Framework, which decouples modality-shared features from modality-specific ones, achieving a 22.5% mAP improvement on the SYSU-MM01 dataset. Similarly, Labbaki et al. innovatively designed an Orthogonal Sequential Fusion Mechanism that mitigates feature redundancy through phased fusion and orthogonal constraints, outperforming traditional parallel fusion approaches. These studies underscore the critical role of inter-modal complementarity mining and structured fusion strategies in enhancing detection robustness.
Self-supervised learning has also achieved breakthroughs in unlabeled data utilization and domain adaptation. Zhang et al. introduced a Pseudo-Label Cross-Teaching Framework for scene adaptation, leveraging background invariance enhancement and hybrid data augmentation to significantly improve detection accuracy in fixed-view scenarios. In monocular depth estimation, Saunders et al. developed a Direction-Aware Convolutional Network that addresses orientation sensitivity through environmental context accumulation, achieving state-of-the-art performance on benchmarks like KITTI. Concurrently, Han et al. constructed a Weather-Augmented Robustness Framework using pseudo-supervised losses and weather-aware augmentation, surpassing existing methods in foggy/nighttime scenarios. These advancements highlight the potential of self-supervised learning for generalization in complex transportation environments.
These additions explicitly connect our work to cutting-edge trends while providing theoretical grounding for future technical developments. The updated content is now integrated into Sections 2.4 and Table 1.
Comment 3: In the experimental section, the author only compared it with old versions, such as YOLOv5/V7, and did not compare it with specialized models in the transportation field.
Reply: Thank you for this critical observation. We have addressed this limitation by adding comprehensive comparisons with specialized transportation detection models in Section 4.2 (see Tables 4 and 5):
KITTI Dataset: Comparisons with state-of-the-art road object detectors (e.g., PSNS-SSD [2024], PointRCNN [2025]) are provided in Table 4. CityPersons Dataset: Miss rate comparisons against occlusion-optimized models (e.g., FA-Net [2022], FCOS w [2024], CCFA-Net [2023]) are detailed in Table 5. Our results demonstrate RE-YOLOv5's competitive advantages in both detection accuracy (+7.63% mAP over PSNS-SSD) and robustness under occlusion (9.68% lower miss rate than CCFA-Net). We apologize for the oversight in not adequately discussing recent specialized models in transportation detection over the past three years, which may have caused confusion. The revised text in Section 4.2 now explicitly highlights these comparisons to better contextualize our contributions.
Comment 4: In the experimental section, a single publicly available dataset was used, which did not include complex urban traffic scenarios.
Reply: We appreciate the reviewer’s concern. However, our experiments are conducted on two public datasets—KITTI and CityPersons—as detailed in Section 4.1.1. Both datasets represent complex urban traffic scenarios, with a specific focus on detecting densely occluded objects. To rigorously evaluate performance:
For KITTI, we statistically analyzed occlusion levels and categorized detection difficulty. Table 4 presents detection accuracy across three object categories (car, pedestrian, cyclist) under varying occlusion ratios.
For CityPersons, we partitioned objects by visibility ratios. Table 5 demonstrates a 9.65% reduction in log-average miss rate for heavily occluded pedestrians compared to baselines.
Furthermore, visualization experiments highlight RE-YOLOv5’s capability to detect severely occluded objects in dense traffic scenes (see Fig. 5 and 6). These results collectively validate our method’s robustness in urban occlusion scenarios.
Comment 5: The introduction of the CBAM module only compares mAP through ablation experiments without analyzing its differential effects on different categories of targets (such as traffic lights and pedestrians).
Reply: Thank you for your attention to attention mechanism analysis. However, our paper does not employ the CBAM module. Instead, we propose a novel Receptive Field Atrous Enhancement (RFAE) module, specifically designed for occlusion handling in traffic scenarios. Unlike CBAM’s decoupled spatial-channel attention, RFAE integrates Atrous Convolutions with multi-scale context fusion to better address dense occlusion detection (as detailed in Section 3.3). The differential impacts of RFAE on various object categories (car, pedestrian, cyclist) are quantitatively analyzed in Table 7.
Comment 6: It is suggested that the author conduct some necessary discussions on hardware deployment costs and model lightweight (such as pruning), which can guide engineering practice.
Reply: Thank you for the valuable suggestions regarding hardware deployment and model lightweighting. While current experimental conditions limit our ability to conduct full hardware deployment tests, we have incorporated critical deployment considerations into our analysis:
Parameter Efficiency: RE-YOLOv5 achieves a parameter count of 49M, comparable to YOLOv5’s 47M, ensuring minimal overhead for edge deployment.
Real-Time Performance: With a speed of 43 FPS (tested on a 3090 GPU), our model meets real-time detection requirements for traffic scenarios. These metrics are now included in the ablation studies (Section 4.4).
Future work will focus on compression techniques such as model pruning or knowledge distillation to further optimize deployability. We will also explore hardware-aware quantization for embedded systems like NVIDIA Jetson platforms.
Reviewer 2 Report
Comments and Suggestions for Authors
- Need to change title. My suggesion: RE-YOLOv5: Enhancing Occluded Road Object Detection via Visual Receptive Field Improvements.
- The abstract should mention quantitative improvements.
- Consider organizing related works into a table for comparison of key features.
- Figure 1, 2, and 3 are helpful—consider adding example feature maps showing the effect of each module (CSPD, RFAE).
- Provide quantitative intuition for EIOU-NMS. Perhaps a small toy example showing how it avoids suppression better than DIOU.
- Rename “Optimization of Non-Maximum Supperssion” to “EIOU-NMS: Enhanced Post-Processing” for clarity and spelling.
- Mention the training epochs, learning rate, optimizer used.
- Inference Time Trade-offs. Include FPS comparisons between YOLOv5 and RE-YOLOv5 across datasets.
- Replace phrases like “boxs” with correct plural “boxes” throughout.
- Consistency. sometimes it’s “RE-YOLOv5,” other times “RE-YOLOV5.” Standardize.
- Clean up long, redundant sentences to improve readability, especially in the introduction and related work.
Author Response
In the PDF file 'Highlighted Manuscript.pdf' within the compressed archive, the modified sections and critical content have been highlighted using color annotations to facilitate review.
Comment 1: Need to change title. My suggesion: RE-YOLOv5: Enhancing Occluded Road Object Detection via Visual Receptive Field Improvements.
Reply: We sincerely appreciate your valuable suggestion on refining the paper title. As advised, we have updated the title to: RE-YOLOv5: Enhancing Occluded Road Object Detection via Visual Receptive Field Improvements.
This revision better reflects our method’s focus on receptive field optimization for occlusion handling. The updated title now appears in the manuscript and LaTeX source files.
Comment 2: The abstract should mention quantitative improvements.
Reply: We sincerely appreciate your suggestion to strengthen the abstract's quantitative focus. The revised abstract now explicitly states:
" The proposed method achieves a mean Average Precision (mAP) of 82.04% on KITTI, representing an improvement of 2.34% over the baseline model. For heavily occluded objects on CityPersons, the Log Average Miss Rate (MR-2) is reduced to 40.31%, a decrease of 9.65% compared to the baseline. These results demonstrate that the proposed method significantly outperforms other comparative algorithms in detecting occluded objects across both datasets."
Comment 3: Consider organizing related works into a table for comparison of key features.
Reply: Thank you for your critical suggestion on organizing related works. We have restructured the comparative analysis of prior studies into Table 1 in Section 2. This tabular format enables readers to quickly grasp the technical evolution and our method’s positioning within the field.
Comment 4: Figure 1, 2, and 3 are helpful—consider adding example feature maps showing the effect of each module (CSPD, RFAE).
Reply: We sincerely appreciate your suggestion to enhance the visualization analysis. As recommended, we have added feature heatmap comparisons in the revised manuscript (see Figure 8 in Section 4.3) to intuitively illustrate the mechanisms and effectiveness of the CSPD and RFAE modules. These visualizations demonstrate how CSPD adaptively focuses on occluded regions via deformable convolutions and how RFAE enhances multi-scale context aggregation through dilated receptive fields.
Comment 5: Provide quantitative intuition for EIOU-NMS. Perhaps a small toy example showing how it avoids suppression better than DIOU.
Reply: Thank you for the valuable suggestion to clarify EIOU-NMS’s advantages. We acknowledge that DIOU (Distance-IOU) is primarily a loss function rather than a suppression criterion. To quantitatively illustrate EIOU-NMS’s superiority over DIOU-based NMS in occlusion scenarios, consider the following example:
Scenario: Two adjacent trucks with overlapping predictions:
Box A (True Positive): Aspect ratio = 3.0, confidence = 0.92 ​Box B (False Positive): Aspect ratio = 1.5, confidence = 0.95 ​Centroid distance = 0.5 pixels, IOU = 0.85 (threshold τ = 0.5). Results:
DIOU-NMS: Penalizes via centroid distance: Score=0.85−0.005=0.845>τ → Incorrectly suppresses Box A.
EIOU-NMS: Jointly penalizes centroid distance and aspect ratio mismatch: Score=0.85−0.005−0.18=0.665<τ.
Correctly retains Box A and suppresses Box B.
This example demonstrates that EIOU-NMS resolves false suppression caused by partial overlaps by jointly optimizing centroid distance and aspect ratio compatibility.
Comment 6: Rename “Optimization of Non-Maximum Supperssion” to “EIOU-NMS: Enhanced Post-Processing” for clarity and spelling.
Reply: Thank you for your suggestion on improving the section title. We have revised the heading of Section 3.4 to "EIOU-NMS: Enhanced Post-Processing" to better reflect its technical focus and ensure terminological clarity. This adjustment aligns with the revised manuscript’s emphasis on occlusion-aware suppression mechanisms.
Comment 7: Mention the training epochs, learning rate, optimizer used.
Reply: Thank you for emphasizing the importance of experimental reproducibility. We have added comprehensive training configurations and hardware/software details in Section 4.1.2:
Training Parameters: Epochs: 300, Learning Rate: 0.01, Optimizer: SGD, Training Strategy: Freezing training with pre-trained weights (backbone parameters fixed during the freezing phase; all parameters optimized during the unfreezing phase).
Hardware: GPU: NVIDIA GeForce RTX 3090 (24GB VRAM), CPU: Intel® Core™ i5-12500 @ 2.50GHz ,OS: Windows 11.
Software: Python 3.9, PyTorch 2.0.1, CUDA 11.3.
These specifications ensure full reproducibility of our experiments.
Comment 8: Inference Time Trade-offs. Include FPS comparisons between YOLOv5 and RE-YOLOv5 across datasets.
​Reply: Thank you for emphasizing the importance of inference efficiency analysis. Based on Table 6, we provide the following quantitative comparisons between RE-YOLOv5 and baseline methods (including YOLOv5 variants).
RE-YOLOv5 Efficiency: FPS: Achieves 43 FPS on KITTI. Parameter Size: 49.334 MB (Method 6), marginally higher than YOLOv5’s 47 MB (Method 1). The full analysis is now integrated into Section 4.3.
Comment 9: Replace phrases like “boxs” with correct plural “boxes” throughout.
Reply: Thank you for your meticulous proofreading. We sincerely apologize for the oversight and have thoroughly revised all instances of "boxs" to the correct plural form "boxes" throughout the manuscript. Additionally, we have verified terminological consistency for other key terms to ensure linguistic accuracy.
Comment 10: Consistency. sometimes it’s “RE-YOLOv5,” other times “RE-YOLOV5.” Standardize.
Reply: Thank you for highlighting this critical terminology inconsistency. We have standardized all instances of the model name to "RE-YOLOv5" throughout the manuscript, including revisions to Sections 2.4, 3.1–3.4, and 4.3, as well as all figures and tables. This ensures consistency in capitalization and hyphenation. We have also performed a full-text verification to eliminate other potential inconsistencies.
Comment 11: Clean up long, redundant sentences to improve readability, especially in the introduction and related work.
Reply: Thank you for your constructive feedback on improving readability. We have systematically streamlined the Introduction and Related Work sections.
Adding logical connectors (e.g., "however," "therefore") to enhance paragraph cohesion; Removing redundant explanations ; Replacing passive voice with active voice where appropriate. These revisions enhance the readability of the article while preserving its academic rigor.
Round 2
Reviewer 1 Report
Comments and Suggestions for Authors The authors have revised the first comments. I have a few small suggestions that need further modification.
1. This paper on road detection is very useful, not only for assisting driving, but also for road maintenance and upkeep. Please further discuss and refer to 10.48130/DTS-2023-0021.
2. This paper involves a lot of symbols, please add a symbol variable explanation table.
Author Response
The authors have revised the first comments. I have a few small suggestions that need further modification.
Comment 1: This paper on road detection is very useful, not only for assisting driving, but also for road maintenance and upkeep. Please further discuss and refer to 10.48130/DTS-2023-0021.
Reply: Thank you for highlighting the broader applications of our road detection framework. We have expanded the discussion in the Introduction (Section 1, Paragraph 3) and incorporated the recommended reference (10.48130/DTS-2023-0021). Key additions are highlighted in yellow in the revised manuscript.
Comment 2: This paper involves a lot of symbols, please add a symbol variable explanation table.
Reply: Thank you for your suggestion to clarify the symbols. We have added Table 2 in Section 3.1 to comprehensively explain all key symbols.
Reviewer 2 Report
Comments and Suggestions for Authors
No need more comments
Author Response
No need more comments
Reply: We sincerely appreciate the reviewers’ feedback and confirmation that no additional comments are required. Based on the finalized review, we hereby confirm:
​​All prior revisions have been thoroughly addressed in the manuscript. The attached clean copy and response letter are formatted per journal guidelines.